# Concrete-to-Abstract Goal Embeddings for Self-Supervised Reinforcement Learning

## Abstract

Self-supervised reinforcement learning (RL) aims to train agents without pre-specified external reward functions, enabling them to autonomously acquire the ability to generalize across tasks. A common substitute for external rewards is the use of observational goals sampled from experience, especially in goal-conditioned RL. However, such goals often constrain the goal space: they may be too concrete (requiring exact pixel-level matches) or too abstract (involving ambiguous observations), depending on the observation structure. Here we propose a unified hierarchical goal space that integrates both concrete and abstract goals. Observation sequences are encoded into this partially ordered space, in which a subset relation naturally induces a hierarchy from concrete to abstract goals. This encoding enables agents to disambiguate specific states while also generalizing to shared concepts. We implement this approach in the context of spatial abstraction. We use a recurrent neural network to encode sequences and an energy function to learn the partial order, trained end-to-end with contrastive learning. The energy function then allows to traverse the induced hierarchy to vary the degree of spatial abstraction. In experiments on navigation and robotic manipulation, agents trained with our hierarchical goal space achieve higher task success and greater generalization to novel tasks compared to agents limited to purely observational goals.

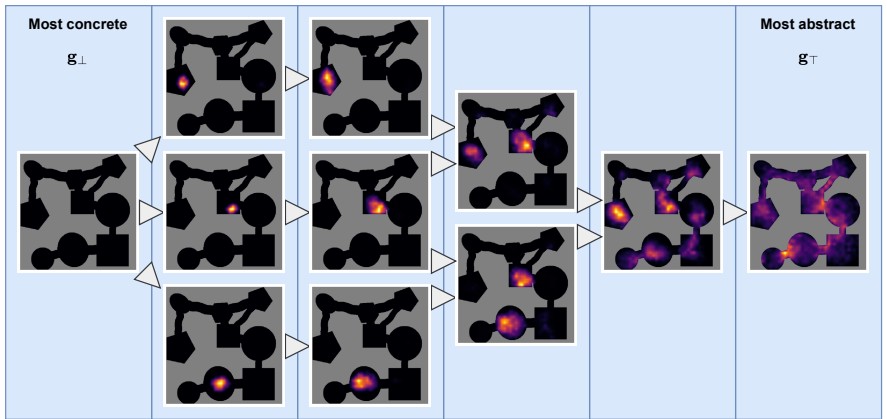

Figure 1: Making goals more abstract in the GridWorld environment.

## 1 Introduction

Over the last decade, reinforcement learning (RL) has achieved remarkable successes, both in mastering highly complex games and in domains where environments provide clearly defined reward functions (Mnih et al., 2015; Silver et al., 2016; Vinyals et al., 2019; Hafner et al., 2025). In real-world applications, however, such reward functions are rarely available or are highly non-trivial to specify (Amodei et al., 2016; Russell, 2016; Christiano et al., 2017). In contrast, humans learn about the world through exploration, play, and observation, largely without external reward signals telling

them what to do (Begus et al., 2014; Gopnik et al., 1999; Goupil et al., 2016). Another problem for real-world applications is task-specificity: an agent trained to solve one task may not be able to apply its learned skills to a different, even slightly modified task (Zhang et al., 2018; Delfosse et al., 2025). One approach to address both the lack of external rewards and the need for generalization is unsupervised RL, in which agents are pretrained without relying on designed reward signals and later adapted to downstream tasks. This pre-training can enable faster inference of policies that generalize across a broad range of tasks. *In this paper, we tackle the problem of hierarchical goal representations for unsupervised (pre-)training within the framework of goal-conditioned reinforcement learning.*

**Unsupervised RL** is characterized by a spectrum of diverse techniques, including intrinsic motivation approaches based on novelty, learning progress, or empowerment (Salge et al., 2014; Zhang et al., 2021), latent skill learning (Eysenbach et al., 2019; Sharma et al., 2020), goal-conditioned RL with self-selected goals (Nair et al., 2018; Bae et al., 2025), approaches that approximate long-term dynamics with successor measures (Agarwal et al., 2025b;a), and contrastive RL methods (Laskin et al., 2020; Schneider et al., 2021; Eysenbach et al., 2022).

**Goal-conditioned RL** (Kaelbling, 1993), extends classical RL by conditioning rewards, policies, and value functions on goals, that effectively serve as an index for multiple tasks that require different desired behaviors. Here, we focus on goal-conditioned RL within a self-supervised learning setting where tasks are specified by the goals the agent attempts to achieve without providing explicit external rewards. This raises the question of how to represent goals alternatively to rewards. A prominent example approach is contrastive reinforcement learning (Eysenbach et al., 2022), where hindsight learning (Andrychowicz et al., 2017) is harnessed to relate current states and actions to future outcomes that are treated as goals. This is achieved by contrastively learning a value function that has high values for matching pairs of state, actions, and outcomes and low values for random pairings.

**Goal representations** in the literature are often expressed directly in observation space (Ghosh et al., 2021; Eysenbach et al., 2022), which ties the agent's capabilities to the abstraction level as well as the structure of the underlying observation space. Typically, environments either provide local observations like egocentric images, or global observations such as precise states and positions. These modalities inherently bias how goals are interpreted: local observations may be ambiguous, leading to abstract goals unless additional context is provided, while global observations specify concrete states but make it harder to capture higher-level abstractions through composition. Thus, neither local nor global observations alone provide a sufficient basis for representing goals, as each is biased towards a single level of abstraction. While the most concrete goals can be seen as elements of a sample space composed of observations or sequences of observations, more abstract goals can instead be seen as (soft) partitions on this space. In general, such goals, that consist of multiple observations, can be induced by constraints (Colas et al., 2019), by utility functions (Christiano et al., 2017; Vamplew et al., 2024) or by desired distributions over observational states (Pong et al., 2020; Ziebart et al., 2008). However, if different levels of abstraction are defined by separate functions or constraints (Sutton et al., 1999; Ho et al., 2019), then it might be difficult to capture their relationship to lower-level goals. An alternative is to represent both concrete and abstract goals in a shared latent space, analogous to word embeddings, where both tokens and longer texts can be encoded in the same vector space (Mikolov et al., 2013).

In the following, we thus propose to study hierarchical latent goal spaces with superimposed partial orders, where all goals are represented as vectors in the same space organized according to varying levels of abstraction by join and meet operations that make goals more or less abstract within the hierarchy. In particular, the partial order is learnt by an asymmetric energy function (LeCun et al., 2006; Grathwohl et al., 2020) that indicates whether one goal forms a subset of another goal, and therefore is a more concrete instance of the same goal, or not. We test this idea in the context of spatial abstraction with goals that encode observation sequences through contrastive training of an asymmetric energy function that indicates whether one observation sequence is contained within another.

The paper is organized as follows. In Section 2 we introduce our methods, including the latent space encoding, the energy function training and the evaluation methods. For evaluation after pretraining, we confront agents with multiple novel reward functions and we search for the best fitting goal in the trained latent space to represent these reward functions. In Section 3, we illustrate our method

in three different RL environments (GridWorld with LiDAR, MemoryMaze, FetchPush). We first illustrate the best fitting goals to represent abstract reward functions, and then show the performance of the corresponding pretrained goal-conditioned policies when searching for the best-fitting goal. In Section 4 we discuss our approach in the context of the wider literature and conclude in Section 5.

## 2 METHODS

**Hierarchical goal spaces organized by subset relations.** Abstraction can be naturally understood as a subset relation: a concrete goal (e.g. this corner) may be considered as subset of a set of concrete goals (e.g. blue corners), which may in turn be a subset of a larger set of subsets (e.g. all corners), etc. where each subset shares some similarity. Of course, there may also be sets of goals that share an intersection, but are not contained within each other (e.g. corners and blue things). Mathematically, the subset relation is a partial order, because it is reflexive, antisymmetric, and transitive. The key idea is to arrange goals according to this partial order, forming a lattice that runs from the most specific to the most general goal. We formalize this idea through a shared latent goal space, $\mathcal{G}$, which supports both representing goals at varying degrees of abstraction and traversing between abstraction levels. Formally, we capture different levels of abstraction by imposing a partial order $\preceq$ over the latent goal space: If $\mathbf{g}_a \preceq \mathbf{g}_b$, then $\mathbf{g}_b$ represents a more abstract goal than $\mathbf{g}_a$. The most concrete goals are denoted by $\mathbf{g}_\perp$, and the most abstract by $\mathbf{g}_\top$, such that

$$\mathbf{g}_\perp \preceq \mathbf{g} \preceq \mathbf{g}_\top \quad \forall \mathbf{g} \in \mathcal{G}.$$

Using the lattice structure of this partial order allows to traverse the latent space through join, $\mathbf{g}_a \vee \mathbf{g}_b = \inf\{\mathbf{g} \mid \mathbf{g}_a \preceq \mathbf{g}, \mathbf{g}_b \preceq \mathbf{g}\}$ and meet operations, $\mathbf{g}_a \wedge \mathbf{g}_b = \sup\{\mathbf{g} \mid \mathbf{g} \preceq \mathbf{g}_a, \mathbf{g} \preceq \mathbf{g}_b\}$. Intuitively, the join identifies the least abstract goal that encompasses both $\mathbf{g}_a$ and $\mathbf{g}_b$, while the meet identifies the most concrete goal they share.

**Energy function and subset relation.** Similarly to how total orders can be represented as unary utility functions under certain conditions (Von Neumann & Morgenstern, 1947; Debreu, 1954), there are multiple equivalent ways to represent partial orders, such as multi-utilities (Evren & Ok, 2011), injective monotones (Hack et al., 2022), or the characteristic function $\chi_\preceq$ of the partial order directly. Specifically, we can model the latter as an "energy" $E_\theta(\mathbf{x}, \mathbf{y})$ that estimates whether $\mathbf{x} \preceq \mathbf{y}$, i.e.

$$E_\theta(\mathbf{x}, \mathbf{y}) \approx \chi_\preceq(\mathbf{x}, \mathbf{y}) = \begin{cases} 1 & \mathbf{x} \preceq \mathbf{y} \\ 0 & \text{else.} \end{cases}$$

While this type of representation does not inherently enforce order properties such as transitivity, it is sufficiently general to learn any relation given enough data. In practice, however, we do not expect to perfectly approximate the characteristic function. Instead, our approach views the order structure as an inductive bias, yielding a hierarchically organized latent space while retaining flexibility to deviate from a strict order. Approximating the characteristic function, however, offers a clear operational advantage: it enables traversal of the induced hierarchy through optimization, holding one input fixed while updating the other. More precisely, we can move upward to find a more abstract goal ($\uparrow$) by maximizing $E_\theta$ over the second entry while keeping the first fixed, or move downward to obtain a more concrete goal ($\downarrow$) by maximizing over the first entry while keeping the second fixed. In particular, given a set of goals $\{\mathbf{g}_i\}_{i=1}^n$, we can find a new goal that is either more abstract or more concrete than the entire set through gradient ascent:

$$
\begin{aligned}
\mathbf{g}_\uparrow^{(t+1)} &= \mathbf{g}_\uparrow^t + \eta \nabla_{\mathbf{g}_\uparrow}\Big|_{\mathbf{g}_\uparrow = \mathbf{g}_\uparrow^t} \sum_{i=1}^n E_\theta(\mathbf{g}_i, \mathbf{g}_\uparrow) \quad &\approx \mathbf{g}_1 \vee \cdots \vee \mathbf{g}_n \quad \textit{(more abstract)} \\
\mathbf{g}_\downarrow^{(t+1)} &= \mathbf{g}_\downarrow^t + \eta \nabla_{\mathbf{g}_\downarrow}\Big|_{\mathbf{g}_\downarrow = \mathbf{g}_\downarrow^t} \sum_{i=1}^n E_\theta(\mathbf{g}_\downarrow, \mathbf{g}_i) \quad &\approx \mathbf{g}_1 \wedge \cdots \wedge \mathbf{g}_n \quad \textit{(more concrete)},
\end{aligned}
\tag{1}
$$

where $\mathbf{g}_\downarrow^0, \mathbf{g}_\uparrow^0 \sim \mathcal{G}$ are initialized randomly to ensure diversity of optimized goals. In Appendix A.1.3, we test the ability of our learned energy model to capture the transitivity of goal sequences generated by Equation 1.

**The case of spatial abstraction.** In a self-supervised setting, the data available to the agent only consists of action–observation sequences without any external rewards. To recover useful relations between observations, the agent could therefore rely on temporal regularities where longer trajectories cover a broader area of the same space and shorter trajectories cover more specific areas of the same space. We therefore propose to treat subset relations between sequences of observations as proxy for the partial order in goal space representing spatial abstraction: if one sequence is contained in another, it corresponds to a more concrete goal within the hierarchy. We use a single contrastive learning framework to learn the encoding $\phi_\theta(\tau)$ of observation trajectories $\tau$ into the latent goal space, as well as the asymmetric energy $E_\theta(\phi_\theta(\tau), \phi_\theta(\tau'))$ which is close to 1 if $\tau$ is a subsequence of $\tau' = (\mathbf{o}_0, \ldots, \mathbf{o}_T)$ and close to 0 otherwise. This enables us to learn both the latent encoding and the ordering relation end-to-end, guided only by temporal consistency in the data.

For practical implementation, we jointly learn the sequence encoding as well as the energy function that induces the partial order, using neural networks trained end-to-end. The overall architecture is shown in Figure 3. To handle sequences of arbitrary length, we use an recurrent neural network (RNN) based encoder (Cho et al., 2014), which maps the sequence $\tau$ into a fixed-dimensional context vector $\mathbf{c}_\tau$. Similar to Hafner et al. (2022), we further encode this context vector into a discrete latent representation $\mathbf{g}$ using a categorical encoder, where the discrete bottleneck encourages the formation of higher-level abstractions over goals. Differentiability is maintained by using a straight-through estimation of the gradients with respect to the discrete samples. This setup is closely related to the discrete variational autoencoder (VAE) used by Hafner et al. (2025), but instead of optimizing a reconstruction loss, we train the model with a reconstruction-free objective based on our proposed subset relation, using contrastive learning.

Given recorded data $\mathcal{D}$ consisting of trajectories $\tau = (\mathbf{o}_0, \ldots, \mathbf{o}_T)$, we construct datasets $\mathcal{D}_+$ and $\mathcal{D}_-$ of positive and negative pairs of trajectories, respectively, based on their relation in sequence space. The energy function $E_\theta$ and sequence encoder $\phi_\theta$ are learned jointly by optimizing

$$
\begin{aligned}
\mathcal{L}(\theta) = &- \mathbb{E}_{(\tau_1, \tau_2) \sim \mathcal{D}_+} \big[ \log E_\theta(\phi_\theta(\tau_1), \phi_\theta(\tau_2)) \big] \\
&- \mathbb{E}_{(\tau_1, \tau_2) \sim \mathcal{D}_-} \big[ \log(1 - E_\theta(\phi_\theta(\tau_1), \phi_\theta(\tau_2))) \big].
\end{aligned}
\tag{2}
$$

Here, $\mathcal{D}_+$ contains pairs $(\tau_1, \tau_2)$ such that $\tau_1 \subset \tau_2$, including cases where both come from the same trajectory as well as composite trajectories formed from unrelated segments. In contrast, $\mathcal{D}_-$ contains pairs where $\tau_1 \not\subset \tau_2$, for example pairs $(\tau_1, \tau_2)$ with $\tau_2 \subset \tau_1$, as well as non-overlapping trajectories, either from the same or from distinct base trajectories $\tau$. Examples are shown in Figure 2. For the full algorithm, see Algorithm 1.

---

**Algorithm 1:** Construction of $\mathcal{D}_+$ and $\mathcal{D}_-$ for contrastive learning

**Input:** Set $\mathcal{D}$ of observation trajectories, batch size $N$

Let $\mathcal{D}_+ = \{\}, \mathcal{D}_- = \{\}$

**for** $N$ *times* **do**

    Sample $\tau_1, \tau_2, \tau_3 \sim \mathcal{D}$ s.th. $\tau_1 \subset \tau_2$

    $\mathcal{D}_+ = \mathcal{D}_+ \cup \underbrace{\{(\tau_1, \tau_2)\}}_{\text{Sub-trajectories}} \cup \underbrace{\{(\tau_1, \tau_1 \oplus \tau_3)\}}_{\text{Composite trajectories}} \cup \underbrace{\{(\{\mathbf{o}\}, \tau_1) \mid \mathbf{o} \in \tau_1\}}_{\text{Individual states}}$

    where $\oplus$ denotes concatenation along time dimension.

**for** $N$ *times* **do**

    Sample $\tau_1, \tau_2 \sim \mathcal{D}$ and $\tau, \tau' \subset \tau_1$, s.th. $\tau \cap \tau' = \emptyset$

    $\mathcal{D}_- = \mathcal{D}_- \cup \underbrace{\{(\tau_1, \tau_2)\}}_{\text{Different trajectories}} \cup \underbrace{\{(\tau, \tau')\}}_{\text{Disjoint segments of same trajectory}} \cup \underbrace{\{(\tau_4, \tau_3) \mid (\tau_3, \tau_4) \in \mathcal{D}_+\}}_{\text{Inversion of positive pairs}}$

**Output:** $\mathcal{D}_+, \mathcal{D}_-$

---

**Abstract hindsight relabeling.** To learn policies in our approach, we chose contrastive reinforcement learning (CRL) which is based on hindsight relabeling of achieved goals (Eysenbach et al., 2022). In standard CRL, relabeling is restricted to single observations from past experience. This limits the diversity of goals available for training, since the agent repeatedly encounters only a narrow subset of the goal space, an effect that becomes more pronounced when training on a fixed, concrete task. Our goal encoding addresses this limitation by allowing entire sequences of observa-

tions to be relabeled as a single, more abstract goal. This provides the agent with richer supervision: it can learn from both fine-grained, single-observation goals and higher-level, temporally extended goals. Moreover, in 10% of the cases we replace the relabeled sequence goal by a more abstract goal by further optimizing our energy function according to Equation 1. We hypothesize that this broader and more expressive goal set improves generalization and task performance. This is supported by an ablation study that can be found in the Appendix, where dropping the optimization step for further abstraction leads to a sizeable decrement in performance. In practice, we randomly sample the sequence length for relabeling, allowing the agent to experience both more concrete and abstract goals. For fair comparison, we also use the same total number of goals for training each algorithm.

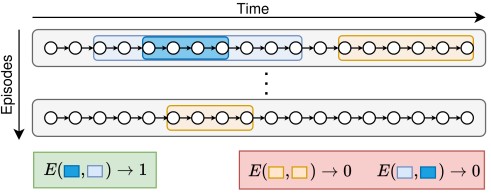

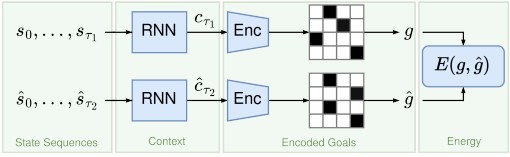

Figure 2: Subset selection for contrastive learning. Positive (green) and negative (red) examples are selected from observation trajectories.

Figure 3: Proposed sequence abstraction model combining a learnable goal representation and similarity measure.

**Abstract hindsight relabeling.** We draw inspiration from zero-shot RL, evaluating agents on novel, unseen reward functions to measure their *persuadability* (Levin, 2022). To compare different goal representations, we first pre-train each agent on the same single-observation goal-reaching task using its respective goal space. After pre-training, each agent explores the environment autonomously to translate previously unseen reward functions into corresponding goals. During this phase, the agent performs fixed-length rollouts and collects rewards while acting under the policy induced by its current best-fitting goal. Importantly, this goal is not static: it is continuously optimized based on the agent's observations and the reward feedback it receives. This setup allows us to assess how effectively each goal representation supports adapting to new reward functions as well as how agents behave under the new goals.

In practice, we employ a hindsight learning procedure that translates rewarding observations into goals. To this end, we minimize the binary cross entropy between rewards and similarities of parametrized $\mathbf{g}_\vartheta$ and achieved goals $\mathbf{g}_t$. More precisely, optimal goals $\mathbf{g}_{\vartheta^*}$ are obtained through

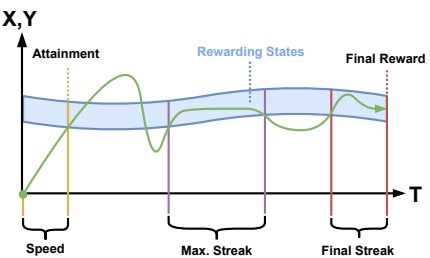

| Criterion | Description |
|---|---|
| Attainment | Rewarding state reached |
| Total Reward | Time fraction at reward |
| Final Reward | Reward in the final state |
| Final Streak | End consecutive rewards |
| Max. Streak | Max consecutive rewards |
| Avg. Streak | Avg. consecutive rewards |
| Speed | Steps to first reward |
| Alignment | Reward-goal similarity |

Figure 4: Agent performance criteria.

$$\vartheta^* = \arg\max_\vartheta \mathbb{E}_{(r_t, \mathbf{g}_t) \sim \mathcal{D}} \left[ \underbrace{-r_t \log \mathrm{Sim}(\mathbf{g}_t, \mathbf{g}_\vartheta)}_{\text{hindsight}} - \underbrace{(1 - r_t) \log(1 - \mathrm{Sim}(\mathbf{g}_t, \mathbf{g}_\vartheta))}_{\text{negative feedback}} \right], \quad (3)$$

where $\mathcal{D}$ are reward-goal tuples obtained from environment interactions and $\mathrm{Sim}$ denotes any similarity measure between goals.

We evaluate the agent's performance given the criteria in Figure 4 that capture different aspects of the learned goal space and policy, such as goal attainment and reward alignment. For the latter we use a fixed set of observations uniformly covering the environment and compute the reward as well as the activity induced by the goal for these observations and compare them in cosine-similarity.

## 3 EXPERIMENTS

We evaluate our approach on different environments with local and global observations. For our approach, observations are encoded into goals using the sequence encoder, and the similarity measure is our energy function, $\text{Sim} = E_\theta$. We compare to a $\beta$-VAE (Higgins et al., 2017; Nair et al., 2018) by encoding observations into latent goals with Euclidean similarity $\text{Sim}(\mathbf{g}, \mathbf{g}') = \exp(-\|\mathbf{g} - \mathbf{g}'\|)$. As a baseline, we use purely observational goals ($\mathbf{g} = \mathbf{o}$) with cosine similarity $\text{Sim}(\mathbf{g}, \mathbf{g}') \propto \mathbf{g}^T \mathbf{g}'$.

Our experiments consist of two phases:

1. **Pre-training:** Policies and goal representations are pre-trained on the original goal reaching task with single observational goals provided by the environment. For our approach, we encode the observations into latent goals. We jointly train policy and goal representation while the agent interacts with the environment.

2. **Evaluation:** We evaluate the frozen pre-tained policies and goal representations on novel reward functions. We find best matching goals to represent the reward functions by optimizing Equation 3. For our approach, we learn a representation of the reward function at hand by optimizing for a more abstract goal than all rewarding goals. During this phase, the agent has to discover the reward function only through environment interactions.

**Goal visualization.** In order to get an intuition of what the goal space actually encodes, we use heatmaps to visualize a particular target goal as follows. First, we sample a set of diverse observations covering possible positions in the environment. Next, we encode them as single-observation goals and compare them to the target goal based on the respective similarity measure. Then we create a heatmap by using the position information and the computed similarities.

### 3.1 ENVIRONMENTS

We use two navigation tasks with ego-centric, local observations. For post-training analysis, we define observation-based rewards (e.g. circular room, yellow corner) as well as spatial rewards (specific room, disjoint rooms). We also use a robot manipulation task with a global, image based observation space with gripper and object-based reward functions. Rewards are binary and state-based. A full list of rewards is given in A.2.

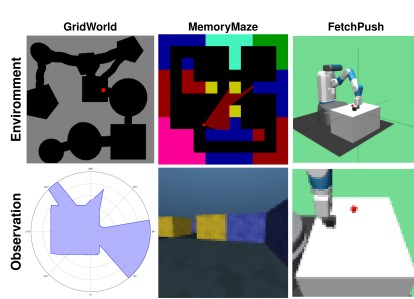

Figure 5: Environments.

**GridWorld** is a 2D navigation task where agents navigate through a grid-based environment using discrete actions. A new episode starts when the agent reaches the goal. Contrary to normal position-based GridWorld environments, our agent receives egocentric LiDAR observations.

**MemoryMaze** (Pasukonis et al., 2022) is originally a long-term memory task based on discrete 2D navigation with image observations. We adapt the task to use the environment for goal-based navigation where goals are provided as image observations. Episodes end when the goal is reached.

**FetchPush** (Plappert et al., 2018) is a robotic manipulation task where a gripper is tasked to move an object to a specific position and keep it there until the episode ends. We use the adapted task from Eysenbach et al. to obtain image-based observations, making the environment challenging. Goal images show the object at some random position with the gripper close to the object. As novel reward functions we define gripper rewards (e.g. moving the gripper to the table border) and object rewards (e.g. pushing the object off the table), where the critical question is whether goals can represent both.

### 3.2 RESULTS

**Traversing the hierarchy.** Optimizing the energy function to traverse the induced hierarchy proves effective for generating goals at varying levels of abstraction. Figure 1 shows how we can

traverse the hierarchy in the GridWorld environment. Starting from the most concrete goal, we can optimize for single-position goals, abstract them to broader regions and even combine individual regions to more abstract, disjoint regions. Finally, the hierarchy is bounded by the most abstract goal. Note that only one possible abstraction is shown at a time, while the partial order allows for different abstractions fulfilling the subset relation. With this observation in mind, we show how we can learn abstract representations of rewards by combining individual goals, corresponding to high rewarding states, into a single abstract goal.

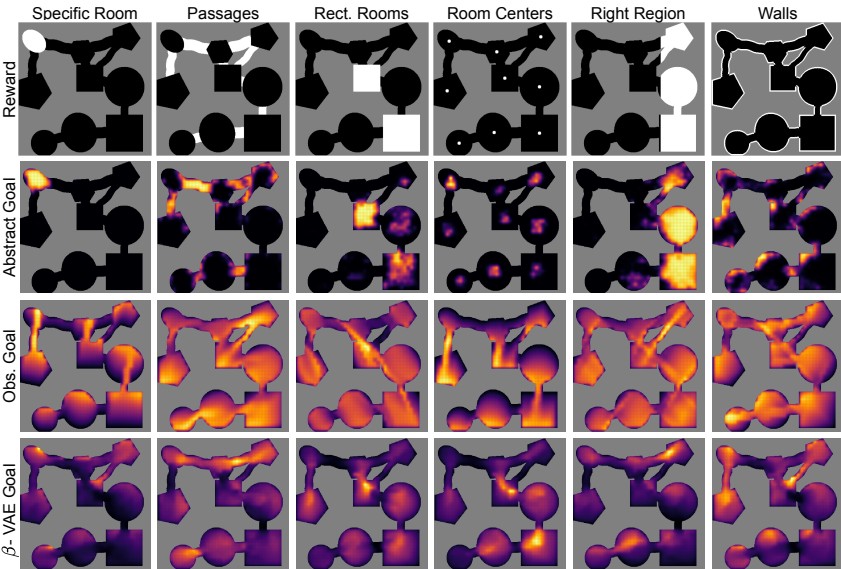

Figure 6: Encoded reward functions in the GridWorld environment. White regions in the top row indicate high rewarding states. There is a single abstract goal to represent each considered reward but no single observation.

**Representing reward functions with goals.** Figure 6 demonstrates our goal representation's ability to encode diverse reward functions in the GridWorld environment, ranging from concrete spatial targets (specific rooms) to increasingly abstract concepts (room centers, disjoint rooms). This spectrum reveals critical differences in representational requirements: while some rewards correspond directly to observable features (e.g., passages that can be identified from single observations), others demand spatial and compositional knowledge of the environment. Our analysis reveals limitations with respect to observational goals. While raw observations can partially encode simple, visually-identifiable objectives like corridor passages, they fail to capture spatial relationships and compositional properties effectively. Furthermore, the reward-optimizing observations show overall high and noisy activity in cosine similarity, indicating fundamental limitations in their capacity to serve as robust abstract goal representations. In contrast, the $\beta$-VAE is able to capture many aspects of the reward functions, although it shows broad activity levels throughout space without clear delimitations due to its distance-based similarity measure. More results supporting our interpretation in the MemoryMaze and FetchPush environments are given in A.1.

**Policy adaptation to novel rewards.** Figure 7 presents a comparative analysis of agents' policies when exposed to the goals that best represent the novel reward functions according to Equation 3. We evaluate the policies according to the performance criteria outlined in Figure 4. Our findings reveal comparable performance of the different agent types in tasks where the available observations are able to capture meaningful aspects of the reward functions. However, a difference emerges for spatial tasks that demand both spatial and compositional knowledge about the environment. Our goal representation demonstrates better performance across most evaluation metrics for spatial rewards, while agents using observational goals exhibit near-random performance, and $\beta$-VAE trained policies are somewhere in between. This performance gap indicates that agents trained with our goal representation have better generalization capabilities for abstract goals that cannot be adequately represented within the observation space. In the FetchPush environment, where all ob-

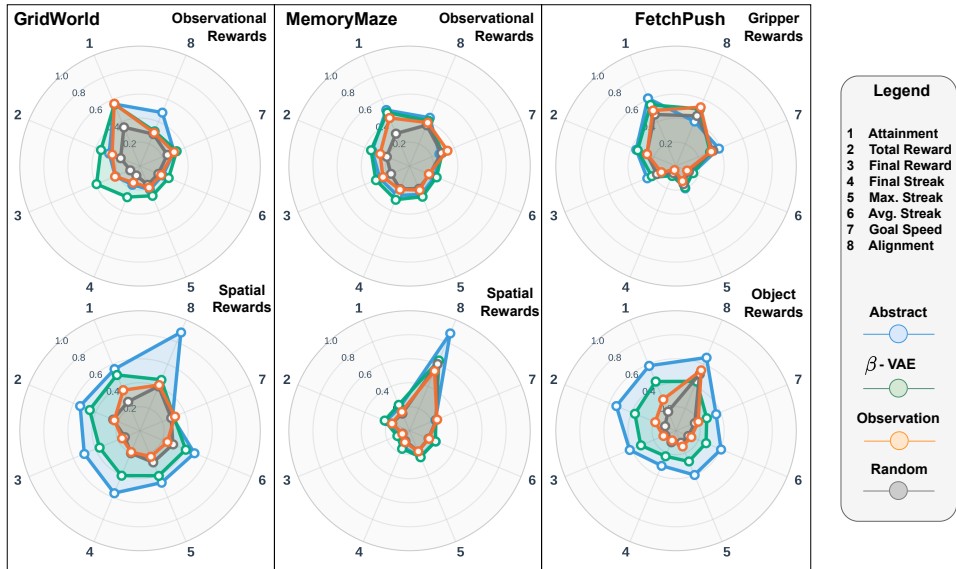

Figure 7: Agent performance on novel rewards for all considered environments.

servations are global, we analyze the agents' behavior on abstract gripper and object-based rewards. We deliberately excluded concrete spatial objectives (such as positioning objects or grippers at specific coordinates) due to their extremely low success probability without sufficient exploration. Our results demonstrate that our goal representation successfully handles both abstract object manipulation and gripper control tasks, and performs significantly better than the $\beta$-VAE trained policy for object-based rewards. In contrast, observational goals fail on both tasks.

## 4 RELATED WORK

**Self-supervised representation learning.** Self-supervised representation learning tries to learn meaningful representations without explicit supervision signals like labels (Ericsson et al., 2021). Contrastive learning has emerged as a powerful paradigm for self-supervised representation learning by augmenting data with labels, often obtained by defining a suitable similarity measure between data points (Jaiswal et al., 2021). The core principle involves bringing semantically similar samples closer together while pushing dissimilar samples apart in the learned embedding space. This approach has shown remarkable success across domains, including computer vision with methods like SimCLR (Chen et al., 2020) as well as natural language processing through approaches like contrastive sentence representation learning (Kim et al., 2021) and supervised contrastive learning for pre-trained language model fine-tuning (Gunel et al., 2021). Temporal contrastive learning extends these principles to sequential data by leveraging temporal relationships as augmentations. Contrastive learning through time (Schneider et al., 2021) takes inspiration from biology to learn object representations by forming augmentations from successive views in temporal sequences. Contrastive learning can be understood through the lens of energy-based models, where the similarity measures used to bring positive pairs together and push negative pairs apart implicitly form an energy landscape over the representation space (LeCun et al., 2006). Such energy functions can be used to learn compositional concepts as shown in Du et al. (2021). Our approach combines the idea of using temporal contrastive learning in combination with an energy function to guide representation learning. Moreover, we extend temporal contrastive learning to not only using temporal relations between individual images but to temporal relations between whole trajectories through the introduced subset similarity in sequence space.

**Abstraction in RL.** The RL literature mainly distinguishes state abstraction and temporal abstraction (Abel, 2020). While temporal abstraction is usually studied in some form of options framework (Sutton et al., 1999), state abstraction reduces the size of the state space by engineering or learning low-dimensional features from raw sensory inputs (Mnih et al., 2015). Both kinds of abstraction can

be conceptualized in terms of subsets, where state abstraction can be understood as a subset structure on the set of states and temporal abstraction as a subset structure on the set of trajectories. Often state abstraction is achieved with reconstruction-based compression methods like VAEs that do not always focus on relevant features (Ha & Schmidhuber, 2018; Hafner et al., 2025; 2019). Hence, there has been a flurry of reconstruction-free compression methods based on contrastive learning (InfoNCE and contrastive predictive coding (Oord et al., 2018; Ma & Collins, 2018), or DeepInfoMax (Hjelm et al., 2019)). Our proposed method for goal embeddings with spatial abstraction follows this line of research, but adds the pre-order structure on the latent space that is absent in previous methods. As we encode observation trajectories, there is also some level of temporal abstraction that we did not explore in the current study. In the future, it will be interesting to pursue this avenue in the context of hierarchical RL (Vezhnevets et al., 2017; Nachum et al., 2018) where concrete and abstract goal vectors could be used to communicate between different agents in the hierarchy. Unsupervised pretraining goal-conditioned policies with such goals could also form the basis to discover a diverse set of skills (Eysenbach et al., 2019).

**Representation learning in self-supervised RL.** The concepts of goals, skills, and intentions share fundamental similarities as they all represent desired outcomes or behaviors that guide agent decision-making. Ghosh et al. (2023) learns intention-conditioned value functions by encoding how outcome likelihoods change when the policy acts with a particular intention in mind. Skill based methods aim to find latent representations of reproducible behavior (Eysenbach et al., 2019; Sharma et al., 2020). To find latent skills, information theoretic ideas are employed to maximize mutual information between states and skills while making skills distinguishable (Eysenbach et al., 2019), between skills and future outcomes (Sharma et al., 2020) or between skills and state transitions (Laskin et al., 2022). Another way to encode diverse behavior is to exploit the linear dependence of the $Q$ value function on the reward Touati & Ollivier (2021); Agarwal et al. (2025b). Agarwal et al. (2025b) show that any agent behavior which can be represented by visitation distributions can be described as a affine combinations of policy independent basis functions. While our current embedding space is based on observational sequences and thus more related to goals, extending this approach to sequences of actions or action and observation sequences may support a representation that forms a bridge between goals and skills.

**Goal-conditioned RL.** Goal-conditioned reinforcement learning enables agents to learn diverse policies by conditioning behavior on desired goals. As the overall objective is attaining a goal, rewards are in general sparse which poses a fundamental problem. Hindsight experience replay (Andrychowicz et al., 2017) and in general hindsight relabeling proved to be an effective method to tackle sparsity by relabeling failed trajectories as success under a different goal Andrychowicz et al. (2017); Ghosh et al. (2021). Different extensions to this idea where proposed to deal with dynamics goals (Ren et al., 2019) or prioritize the experience for better relabeling (Zhao & Tresp, 2018). Ghosh et al. (2021) rephrase the goal-conditioned learning problem as supervised learning without rewards, by relabeling experiences trajectories as success and using self-imitation learning to directly optimize the policy. Eysenbach et al. (2022) demonstrated that contrastive learning can be reinterpreted as goal-conditioned reinforcement learning, showing that contrastive objectives naturally lead to goal-reaching behavior. In our work we simply used existing goal-conditioned RL approaches, in particular contrastive methods, to learn policies. The innovation of our work focuses on the representation of the goal space that can be used by such methods.

**Goal representation learning** While observations are commonly used as goals, various approaches for learning latent goal representations have been proposed. Nair et al. (2018) employ a VAE with reconstruction loss to embed observations into a latent space, enabling the sampling of novel goals and computation of distances in latent space. Building on this foundation, Nair et al. (2020) extend the approach using a conditional VAE that incorporates future goals to encode goal feasibility. Co-Reyes et al. (2018) take a different approach by encoding entire trajectories into latent representations using a VAE trained with reconstruction loss, subsequently training a policy conditioned on these latent trajectories to replicate the encoded sequences. Hafner et al. (2022) utilizes discrete latent sub-goals derived from a discrete VAE applied to world model states to guide reward optimization. We built on both ideas, encoding trajectories into latent representations and using a discrete VAE for encoding goals but in contrast to other approaches, we use reconstruction only to facilitate initial convergence before fully transitioning to contrastive learning.

**Evaluation.** Self-supervised RL methods like goal-conditioned RL and successor feature methods are often used to study zero-shot RL (Schaul et al., 2015; Barreto et al., 2018), i.e. instant generalization to unseen tasks. While we also evaluated our method in terms of adaptation to novel reward functions, our policies where not optimized during pretraining for abstract goals. Instead, we focused on representation learning of the goal space in the energy function and showed that there is some emergent capability of our policies to deal with abstract goals despite being trained exclusively on concrete goals. In the future it will also be interesting to compare our method to other zero-shot RL methods, but for this comparison to be meaningful our policies should be also pretrained with abstract goals, which requires a kind of curriculum, which was not the purpose of the current study.

## 5 CONCLUSION

In this paper, we introduce a novel approach for representing goals as embedding vectors in a latent space with varying levels of abstraction for self-supervised reinforcement learning. Existing methods typically define goals either through hand-engineered, goal-dependent reward functions or directly in terms of observations, thereby constraining the level of abstraction to the properties of the observation space. We demonstrate the feasibility of such unified goal embeddings in the case of spatial abstraction by encoding sequences of observations into a latent goals, learned in an unsupervised manner with contrastive learning, where a partial order naturally induces a hierarchy with spatially broad abstract goals and spatially more focused concrete goals. Traversing this hierarchy in our goal space to unseen goals can be exploited to encode novel reward functions as goals. Through experiments in navigation and robotic manipulation, we have demonstrated that agents trained with our hierarchical goal space achieve higher task success and significantly greater generalization to novel, unseen tasks compared to agents reliant on purely observational goals.

While our goal representation is shown to be effective at encoding a variety of reward functions, there are several remaining challenges. First of all, we only study one particular kind of abstraction, many others are thinkable (e.g. based on feature similarity across trajectories). Second, currently the agent's policy does not see very abstract goals during pre-training, but only abstract goals representing close neighbourhoods. Accordingly, it is not surprising that results show a discrepancy between what the goal representation can encode and what a pre-trained agent can actually achieve with these abstract goals. Therefore, future work should focus on developing a more effective mechanism for training agents to fully utilize these rich, abstract representations. The concepts of compositionality and multi-level abstraction could also be explored further, particularly in the context of hierarchical RL where goal vectors with multiple levels of abstraction could be easily communicated between different modules. Ultimately, our goal representation approach can be used in conjunction with other RL methods in an unsupervised fashion to foster learning and generalization in the absence of explicit reward functions, a critical step toward applying reinforcement learning to open-ended environments.

## REPRODUCIBILITY STATEMENT

We commit to release code upon acceptance. Code will be made available to the reviewers via an anonymous repository. Our implementation includes network architectures and code to learn the proposed goal representation as well as helper scripts for visualization. Hyper-parameters and architectural details are specified in the appendix A.3.

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

# A APPENDIX

## A.1 ADDITIONAL RESULTS

Here we present additional results on the representative power of our goal representation.

### A.1.1 ADDITIONAL GOAL HEATMAPS

First, we consider the MemoryMaze environment where we learn abstract goals from image observations by using interaction with the environment. Our goal representation is able to encode all considered goals, ranging from observational goals like colored corners or walls to more abstract spatial goals combining different rooms as shown in Figure 8. The $\beta$-VAE based approach is able to represent observation goals and most abstract goals while showing overall high activity in similarity. In contrast, observational goals struggle with encoding spatial concepts like a specific room but are able to encode some observational properties like colored walls or corners. For the Fetch-

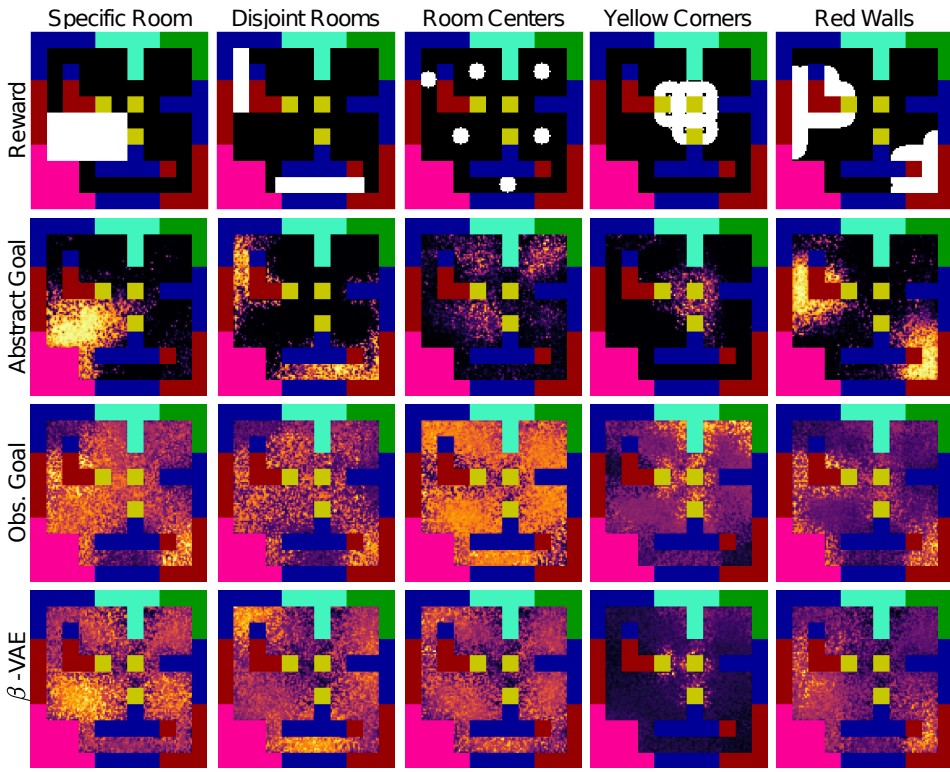

Figure 8: Encoded reward functions in the MemoryMaze environment. White regions in the top row indicate high rewarding states.

Push environment, only spatial goals regarding the gripper and object are considered (see Figure 9). While our method is able to encode both, gripper based (specific regions, off-table) and object based (specific regions, off-table) rewards, the observational goals are not able to capture any meaningful structure. The $\beta$-VAE is able to represent some gripper based rewards but struggles with the object based rewards.

### A.1.2 ABSTRACT GOAL RELABELING

For our goal relabelling during contrastive learning we use entire sequences of observations to be relabeled as a single, more abstract goal. In 10% of the cases we replace the relabeled sequence goal by a more abstract goal by further optimizing our energy function according to Equation 1. This extra optimization allows training the policy with slightly more abstract goals than the ones that were actually experienced. Figure 10 shows the difference in performance of the policies with

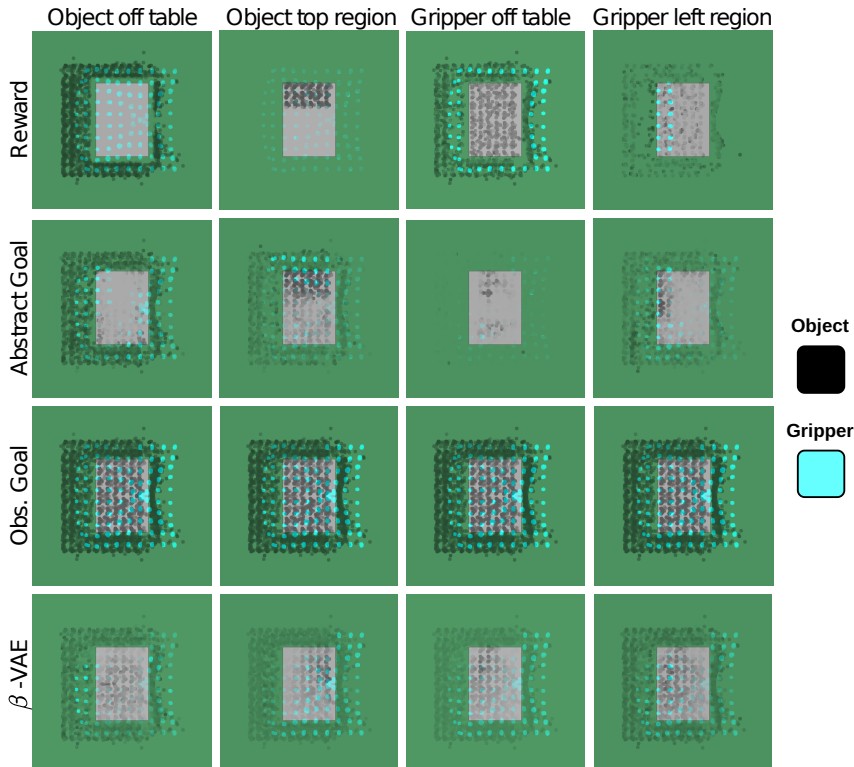

Figure 9: Encoded reward functions in the FetchPush environment (top-down view). Black regions indicate high rewarding states for the object while cyan regions are used for gripper rewards. Only our approach is able to represent both, gripper and object rewards.

and without this extra optimization step. Especially, for spatial and object-based reward functions the extra abstraction also leads to an extra improvement in performance.

### A.1.3 TRANSITIVITY ANALYSIS (GRIDWORLD)

To assess the quality of the abstractions contained in the energy function, we check for internal consistency, in particular transitivity, i.e. if $\mathbf{g}_0 \preceq \mathbf{g}_1$ and $\mathbf{g}_1 \preceq \mathbf{g}_2$ then it should follow that $\mathbf{g}_0 \preceq \mathbf{g}_2$. Starting from a goal $g_0$ grounded in the agent's experience, we generate a sequence $(g_0, g_1, \ldots, g_t)$ of goals iteratively. Subsequent goals are obtained through optimization of the energy following Equation 1, i.e., $g_t = \operatorname{argmax}_g E_\theta(g_{t-1}, g)$. Consequently, subsequent goals do not necessarily correspond to a particular experienced sequence but, for example, might represent a disjoint set of observations that share some common feature (e.g., being close to a wall, center of a room, corners, etc.). Since the energy never experienced these more abstract goals during training, we naturally expect a degradation in goal quality as more steps are taken to obtain more and more abstract goals. We see this degradation directly in the decreasing final energy values achieved through this optimization (Figure 11d), which directly matches the decrease in transitivity (Figure 11a).

In practice, due to the variability of the random initialization in Equation 1, for each goal $g_t$ we generate 50 candidates for $g_{t+1}$ and select the one with the highest optimized energy. We show both the best and the mean energies in Figure 11d. The whole process is repeated 100 times starting from different samples $g_0$. We optimize each goal using Equation 1 for 50 steps with $\eta = 10$. For transitivity to hold, we should find that the energy $E_\theta(\mathbf{g}_t, \mathbf{g}_{t+\tau})$ should be close to 1 for $\tau \geq 0$ and close to zero for $\tau < 0$. Illustrated as a matrix in Figure 11a the elevated energies form a soft lower triangle consistent with the transitivity requirement. In Figure 11b we check whether active regions of more abstract goals subsume the active region of the precedent goals, where perfect subsumption would correspond to a fraction value of one. A state is viewed as active (i.e. part of the goal) when the energy exceeds 0.7, which introduces a source of imperfection. As a comparison, we also show

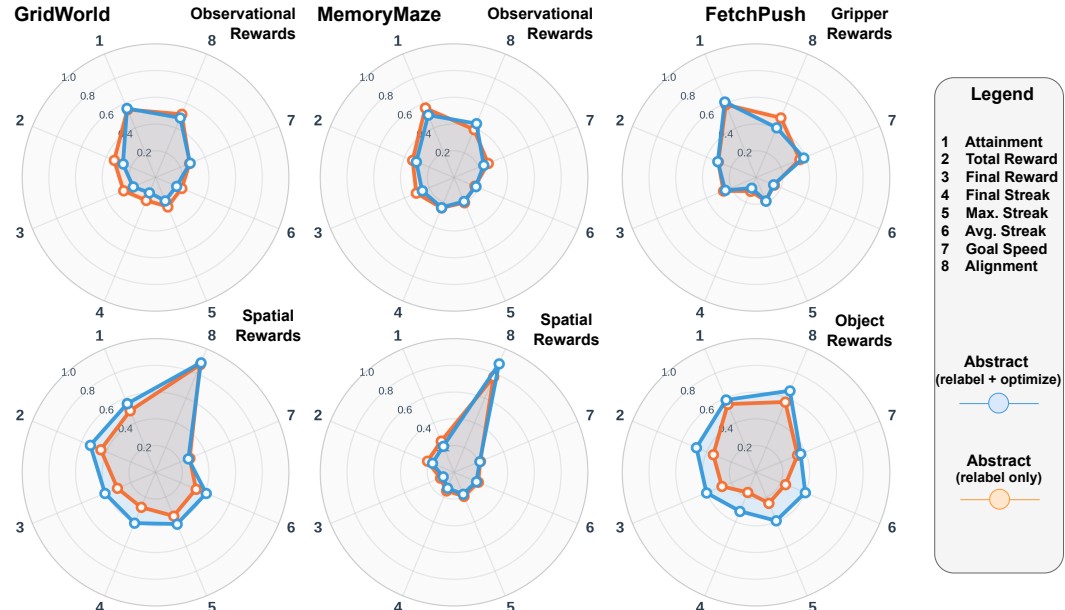

Figure 10: Agent performance on novel rewards for all considered environments using different relabeling strategies for our approach.

the fraction of active states for the same goal when it is sampled 10 times and we count the common activation, as shown by the orange line. The deviation from 1 can be explained by noise in the sampling process.

Figure 11c shows that more abstract goals indeed cover larger areas of space, as would be expected for spatial abstraction. Figure 11d shows both average and maximal energy values across iterations between goals and their super-goals, which would be equal to one in case of perfect optimization of a perfect energy function. We see that the average energy drops with increasing optimization, implying that it becomes more and more difficult to find good abstract goals with gradient optimization, the higher the abstraction level.

## A.2 ENVIRONMENT SPECIFIC REWARDS

We add novel reward function to the GridWorld, MemoryMaze and FetchPush environments. Table 1 shows an overview of all rewards used for evaluation. For the GridWorld and MemoryMaze environments, rewards are categorized as spatial (requiring positional and compositional knowledge) or observational (encodable from single observations). For the FetchPush environment, reward functions focus on spatial properties of the gripper and object as the observations contain solely global information.

## A.3 MODEL ARCHITECTURES & HYPERPARAMETERS

**Contrastive RL.** We base our implementation of CRL (Eysenbach et al., 2022) on the code provided by Zheng et al. (2024). For all experiments we use a two layer multi-layer perceptron (MLP) based policy with 256 hidden units each and ReLU activation. For image based environments, all images are scaled to $64 \times 64 \times 3$. We encode images to latent representation size of 256 using a convolutional neural network (CNN) based encoder as proposed in Mnih et al. (2013). The encoders in the parametrized $Q$-value function consist of MLPs of two hidden layers with 256 units and ReLU activation, projecting down to a representation dimension of 32. Note that the encoder consists of two separate networks for state-action and goal encoding. For experiments using continuous actions we use adaptive entropy regularization as proposed in Haarnoja et al. (2018) with a target entropy of 0.0. For discrete actions we bound the maximum $D_{\mathrm{KL}}$ between policy and a uniform prior policy by 1 via minimization ("free-bits" trick proposed in Kingma et al. (2016) and used by Hafner et al.

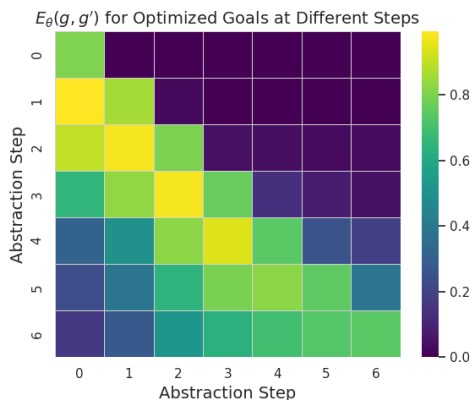

a) Energy activity for all optimized goals, comparing different abstraction steps. A lower triangular matrix indicates correct transitivity where more abstract goals are a superset of preceding goals and asymmetry holds.

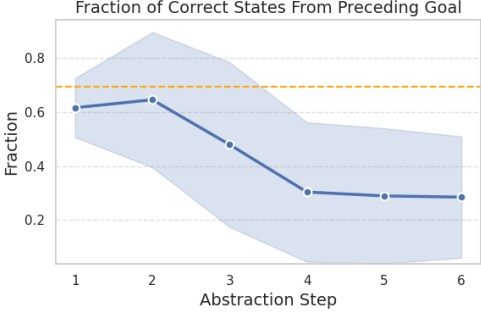

b) Fraction of active states preserved between abstraction steps. An ideal more abstract goal should cover at least the same states as the previous, more concrete goal. Mean and standard deviation are shown. The orange line shows the average fraction of correct states when sampling the same preceding goal multiple times.

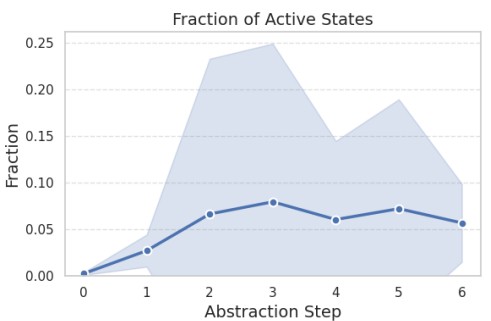

c) Fraction of active states in the environment. More abstract goals yield higher coverage, indicating larger active regions. Mean and standard deviation are plotted.

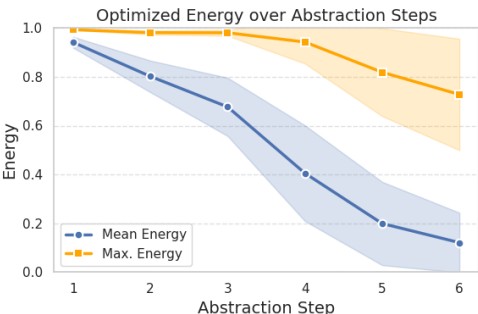

d) Optimized energy over all abstraction steps. Mean energy denotes the average over the 50 rollouts per goal while the maximum energy corresponds to the goal selected for the next abstraction step. Mean and standard deviation are shown.

Figure 11: Analysis of optimized abstract goals in the GridWorld environment.

| Environment | Category | Rewards | Description |
|---|---|---|---|
| **GridWorld** | Spatial | Individual rooms
Disjoint rooms (two rooms)
Regions (top, down, left, right) | Requires spatial knowledge |
| | Observational | Room centers
Shaped rooms (elliptical, polygonal, ...)
Close to wall
Passages (between rooms) | Possible to encode by a single observation |
| **MemoryMaze** | Spatial | Individual rooms
Disjoint rooms (two rooms) | Requires spatial knowledge |
| | Observational | Looking at walls (any wall, red wall)
Looking at colored corners (blue, yellow) | Possible to encode by a single observation |
| **FetchPush** | Gripper-based | Off table
On table border
In table region (top, left, bottom, right) | Focus on gripper representation encoding |
| | Object-based | Off table
On table border
In table region (top, left, bottom, right) | Focus on object representation encoding |

Table 1: Environment-specific reward functions.

(2025)). Batch-sizes vary depending on the problem and are mostly limited by memory: 128 for image based tasks and 2048 for all other experiments. We try to maximize batch sizes as Zheng et al. (2024) showed that in general larger batch sizes are desired for CRL.

**$\beta$-VAE.** As a baseline we train $\beta$-VAEs (Higgins et al., 2017; Nair et al., 2018) to encode observations into latent goals, using a latent dimension of 4 and a $\beta$ of 10. Following Nair et al. (2018), we use Euclidean distance in latent space as similarity measure. We obtain a similarity function $\text{Sim}(\mathbf{g}, \mathbf{g}') = \exp(-\|\mathbf{g} - \mathbf{g}'\|)$ resulting in high similarity for low distance and low similarity for high distances. This way, the similarity is bounded and suitable for encoding reward functions using Equation 3.

**Trajectory sampling for contrastive learning.** For learning our representation using contrastive learning as described in Section 2 we first sample trajectories of length $T \sim \text{Unif}(3, 50)$ from the experience. We then independently sample sub-trajectories of length $\text{Unif}(1, \lfloor T/3 \rfloor)$. This formulation guarantees that the original trajectories of length $T$ can accommodate two non-overlapping sub-trajectories, each with a maximum length of $\lfloor T/3 \rfloor$, separated by a gap of at most $\lfloor T/3 \rfloor$ steps.

**Goal Abstraction.** We encode sequences with a gated recurrent unit (Cho et al., 2014) with a hidden state size of 256. After encoding the whole sequence we use the last hidden state and use a discrete VAE (Hafner et al., 2022; 2025) using two hidden layers to project the RNN hidden state to a multi-categorical distribution. We use 16 categories with 16 possible values, resulting in a one-hot encoding of shape $16 \times 16$. For image based observations we first encode the images using the CNN described in Mnih et al. (2013) to obtain a 256 dimensional encoding and apply our architecture to the encoded images. The energy function is parametrized as a simple 3 layer neural network with ReLU activation and a hidden dimension of 256. We use binary cross entropy as loss to train the energy function and a fixed batch size of 256 for all experiments.

**Reward encoding.** While rewards are translated into goals $\mathbf{g}_\vartheta$ by optimizing (3), we drop the negative feedback term when encoding observational goals in the FetchPush environment, because constrastive learning would eliminate static background information in this case and thereby destroy almost all observational information.

To practically encode rewards into goals we parametrize the goal by a simple neural network, receiving a zero-vector of shape 64 as input and outputting a goal in the corresponding goal space. For image based goals, we use three transposed convolution layers, reversing the architecture of the

CNN encoder. For the LiDAR observations and our approach we use a simple three layer MLP. Discretization for our goals is achieved by a final categorical distribution.

**Network sizes.** We ensured that network sizes are comparable between different goal representations by increasing/decreasing the width of hidden layers accordingly to match the number of trainable parameters.

## A.4 TRAINING PROCEDURE & EVALUATION

We split training into a pretraining phase, where the policy and goal representation is pretrained, and a fine tuning phase, where we optimize over goals to encode reward functions. In the pretraining phase, we collect interactions in the environments for 16 steps while performing one training step as a trade-off between sample efficiency and speed. The agents are pre-trained on the original, single observation goal reaching task where observations are provided by the environment. For GridWorld and FetchPush, pre-training lasts for 1000000 environment steps. For MemoryMaze, we use 500000 environment steps. During the first 50000 environment steps we use additional losses to stabilize training and improve performance. First, we use a reconstruction objective, reconstructing single observations from the encoded goals. This helps with early learning, especially in image based environment where otherwise the subset relation takes a long time to find good representations. Note that we fully transition to contrastive learning later in training, as reconstruction hinders convergence at some point. Furthermore, we regularize the discrete latent space by imposing a $D_{\mathrm{KL}}$ constraint on the categorical distribution effectively regularizing it towards a uniform distribution. This constraint loosens over the course of the first 50000 environment steps. This is required as during early training the encoder may collapse.

We evaluate what rewards each considered goal representation is able to represent as well as how well the pre-trained and frozen policies perform. For each novel reward function, the agent interacts with the environment for a total of 20000 steps. The environment steps are split up into episodes of 50 steps. The resulting behavior is analyzed using the goal representations are evaluated based on heatmaps as well as by the alignment criterion. The agents behaviors is assessed by the performance criteria given Figure 4 and depicted in Figure 7. After learning the reward function, the agent is tasked to reach the optimized goal 100 times starting from different initial states and with a budget of 50 environment steps. For each reward function, three runs are performed and averaged. Then reward groups are averaged to obtain the final plot. The star plot in Figure 7 min-max normalizes the performance metrics such that 0 is the worst observed performance and 1 is the best observed performance over all rewards in an environment. Attainment dependent metrics are scaled by the attainment value such that non-attaining episodes result in worst performance (0).

## A.5 USE OF LLMS

This work utilized large language models (LLMs) for two specific purposes: (1) manuscript preparation, especially grammar checking and LaTeX formatting assistance, and (2) visualization enhancement, where LLMs helped to improve plotting code aesthetics and to generate visualizations for the environments. Importantly, LLMs were used solely for presentation and formatting purposes. All underlying research data and experimental results remain unmodified to preserve scientific integrity.

