# OpenReview forum: "Concrete-to-Abstract Goal Embeddings for Self-Supervised Reinforcement Learning"
_ICLR.cc/2026/Conference — Submitted to ICLR 2026_

### Official Review · Reviewer_Xcv6 · 2025-10-27

**Soundness:** 3
**Presentation:** 3
**Contribution:** 3
**Rating:** 4
**Confidence:** 4

**Summary:**

This paper introduces a method for learning a hierarchical goal space for self-supervised RL. The core idea is to encode entire sequences of observations into a latent space. A partial order is induced on this space by training an energy function to model the subsequence relation between trajectories. The authors posit that this partial order corresponds to a hierarchy of abstraction, where a goal corresponding to a subsequence is more concrete than the goal for the full sequence. This structure is learned end-to-end using contrastive learning. The learned energy function can then be used to traverse the hierarchy via optimization to find more abstract or concrete goals. Experiments in navigation and manipulation tasks show that this representation can capture abstract reward functions and leads to better generalization on novel tasks compared to using single observations as goals.

**Strengths:**

- Originality: The primary strength is the novel idea of using the subsequence relation to induce a partial order and thus a concrete-to-abstract hierarchy in a latent goal space. This is a creative and principled approach to self-supervised structure learning.

- Methodology: The end-to-end learning framework, which combines a sequence encoder with a contrastively trained energy function, is elegant. The ability to traverse the learned hierarchy by optimizing the energy function is a powerful and interesting feature.

- Significance: The work addresses the important problem of goal abstraction in RL. By enabling agents to represent and potentially reason about goals at varying levels of abstraction, this work is a valuable step towards more general and capable autonomous agents. The empirical results clearly show the limitations of standard observational goals for abstract tasks and demonstrate the effectiveness of the proposed representation.

**Weaknesses:**

- Questionable Core Assumption: The paper's foundation rests on the assumption that the subsequence relation is a good proxy for abstraction. This is not always true. A short, specific trajectory (e.g., entering a room and turning left) is concrete, but a very long trajectory covering the entire map might also be viewed as a single, concrete instance of behavior, not an "abstract" goal. The paper does not discuss the limitations of this assumption or scenarios where it might fail.

- Limited Baselines: The experiments almost exclusively compare against using single observations as goals. While this is a common baseline, it is insufficient for evaluating a method designed for abstract goals. Stronger baselines would include other latent goal methods, such as those using VAEs to encode observations [1] or methods that encode full trajectories [2], even if for different purposes. Without these comparisons, the performance gains are hard to contextualize.

- Disconnect between Representation and Policy: As acknowledged in the conclusion, the policy is only trained on concrete, single-observation goals. The rich hierarchical structure of the goal space is learned but not directly used during policy pre-training. This is a significant limitation. The agent is never explicitly taught how to achieve an abstract goal, which likely hampers its ability to fully leverage the powerful goal representation.

- Source code: Anonymous link to the source code is missing despite mentioning in the Reproducibility Statement.


[1] Nair, Ashvin V., et al. "Visual reinforcement learning with imagined goals." Advances in neural information processing systems 31 (2018).

[2] Co-Reyes, John, et al. "Self-consistent trajectory autoencoder: Hierarchical reinforcement learning with trajectory embeddings." International conference on machine learning. PMLR, 2018.

**Questions:**

- Could you elaborate on the limitations of using the subsequence relation as a proxy for abstraction? For example, is a trajectory that visits states (A, B, C) always more concrete than one that visits (A, B, C, D, E)? What if the first trajectory represents a complete, short skill, while the second is a longer, meandering one?

- Why was the comparison limited to single-observation goals? A comparison against other latent goal representation learning schemes (e.g., VAEs on states, trajectory auto-encoders) would significantly strengthen the paper's claims.

- The conclusion mentions that the policy is not trained on abstract goals. Have you run any experiments where the policy is trained or fine-tuned on goals generated by traversing the learned hierarchy (i.e., the outputs of the join operation)? It seems this would be a critical step to fully realize the benefits of your proposed goal space.

- The process for finding abstract goals involves optimizing the energy function starting from a random initialization g^0. How sensitive is the final abstract goal to this initialization? Does the process consistently yield semantically meaningful and diverse abstractions, or does it often get stuck in trivial local optima?

---

> ### Author Response · Authors · 2025-11-20
>
> We thank the reviewer for the constructive comments and pointing out that we have not clearly separated different aspects of abstraction. We revised the paper and adress the concerns and questions below.
>
> 1. Core assumption about abstraction: In the restructured revision, we now start straight-away with our core concept of abstraction as a subset relation, i.e., our main assumption is that abstraction essentially means to combine a collection of entitities into a new entity. Mathematically this is represented through a partial order. We clarify in the paper that here we only deal with one particular kind of abstraction, namely spatial abstraction, constructed from temporal proximity in experienced observation sequences, where longer sequences cover broader area of space and shorter sequences cover smaller areas of space. Crucially, however, there are many other kinds of abstraction that would fit into the more general subset picture.
>
> 2. New VAE baseline: In the revised version, we have added an additional comparison with a VAE-based approach. The revised results show that our approach improves the results on abstract goal tasks, in particular in terms of representing unseen reward functions through single abstract goals.
>
> 3. Policy training with abstract goals: Thank you for this proposal. We have added a new analysis where the policy is also trained on a more general goal than the relabelled goal. This goal is found by gradient ascent of the energy with the relabelled goal being fixed (Eq. 1).
>
> 4. Sensitivity of abstract goals based on different samples of $\mathbf{g^0}$: We thank the reviewer for this question. We have added an evaluation that tests our learned energy and abstraction models for consistency regarding transitivity and coverage across a number of gradient ascent applications starting from different samples of $\mathbf{g}^0$.

---

### Official Review · Reviewer_eU52 · 2025-10-30

**Soundness:** 2
**Presentation:** 2
**Contribution:** 2
**Rating:** 2
**Confidence:** 3

**Summary:**

This paper proposes a new way to learn goals representations in a self-supervised way. Instead of treating goals as fixed observations or states, it builds a hierarchical goal space that moves from concrete to abstract goals. The idea is that concrete goals, such as reaching specific locations or achieving low-level actions, can be summarized or combined into more abstract goals that can represent a more general task. For example, in the GridWorld environment, reaching a room’s corner is a concrete goal, while exploring the entire room forms a more abstract goal that encompasses several of those concrete goals. Sequences of observations are encoded with an RNN and a partial order between goals is learned using an energy-based contrastive loss that determines whether one goal is a subset of another. This structure lets the agent move up or down the hierarchy, discovering broader or more specific goals as needed. This representation is then plugged into a contrastive goal-conditioned RL framework, showing that it helps agents generalize to unseen reward functions and tasks in navigation and robotic manipulation settings.

**Strengths:**

1) The idea of viewing goal spaces as partially ordered hierarchies and modeling them as such is conceptually creative and could inspire future work on abstraction in RL.

2) The aim to unify concrete and abstract goal representations is an important step toward generalization in RL.

3) New evaluation metrics shows effort towards more broad validation (Figure 4 and Figure 7). Figure 4 lists the proposed metrics: Attainment, Total Reward, Final Reward, Final Streak, Max Streak, Average Streak, and Goal Speed. Figure 7 then uses these metrics to compare their proposed abstract goal representation method with observational and random goal baselines. These metrics capture performance evaluation dimensions beyond raw success rate, and offer a more nuanced assessment of generalization.

**Weaknesses:**

1) The paper’s main idea of learning subset relations through an energy function lacks theoretical justification. Energy-based and contrastive learning frameworks are typically used to model equivalence or symmetric similarity relations rather than asymmetric hierarchical ones. The proposed formulation fails to demonstrate how the learned energy function satisfies the properties of a true partial order. Additionally, the paper does not clarify how the model handles pairs of trajectories that have no inclusion relation where neither is a subset of the other. Specifically, during the hierarchy traversal step, the optimization can arbitrarily drift toward unrelated goals, as no mechanism is described to ensure that the gradient of the energy function leads to more concrete yet related goals. This issue is particularly concerning when unrelated goals also yield an energy value of zero effectively breaking the intended hierarchical structure of the learned goal space.

2) Missing definitions (e.g., $g^*$ and $p_\vartheta$ in lines 199–210), inconsistent notation (use of E(⋅,⋅) for both discrete binary ordering and continuous similarity), unnumbered equations, and a confusing flow make the paper difficult to follow. Besides, adding diagrams or pseudo-code could greatly improve clarity by illustrating how the components interact and how the proposed hierarchy is implemented in practice.

3) The experiments lack comparison with baselines (e.g., VAE-based latent goal RL, zero-shot RL methods), making it hard to assess performance improvements. Reported results show modest improvements and lack statistical analysis or ablation.

**Questions:**

1) What guarantees (if any) exist that the energy function truly models a partial order rather than just a similarity metric?

2) In the reward encoding loss, how is $g^*$ obtained and optimized? What is the meaning of $p_\vartheta$?

3) Could you provide a clearer visualization of concrete-to-abstract goals to show the method’s ability to learn all ranges of abstraction?

4) Why are there no comparisons with VAE-based or other latent-goal models? Could you provide additional results comparison with baselines?

---

> ### Author Response · Authors · 2025-11-20
>
> We thank the reviewer for the detailed and constructive comments, which helped us clarify the exposition and strengthen the paper. Below, we address each concern and describe corresponding improvements to the revised manuscript.
>
> 1. Use of the term "energy function": We agree that the term energy function can be ambiguous across learning paradigms. In our work, the energy does not imply symmetry, as in typical similarity-based formulations. Instead, we follow the definition of energy functions as compatibility measures [1]: mappings from pairs of inputs to a scalar value expressing how well they "fit". In our case, this compatibility corresponds to the partial order relation in the latent goal hierarchy. In particular, our case is consistent with the intended use of energies as optimization objectives: we move through our hierarchy by optimizing over one of the inputs while keeping the other fixed. This is similar to how energy-based models are used to implicitly model correspondences between, say, images and labels, where one can do both classification and generation with a single energy model through optimization [1-3]. We have clarified this in the text.
> - [1]: LeCun et al., "A Tutorial on Energy-Based Learning". Predicting Structured Data, Neural information processing systems, 2007
> - [2]: Du et al., "Learning Iterative Reasoning through Energy Minimization". PMLR 2022.
> - [3]: Grathwohl et al., "Your classifier is secretly an energy based model and you should treat it like one". ICLR, 2020
>
>
> 2. On the properties of a true partial order: We appreciate this observation. We have clarified in the new version that multiple function classes can represent partial orders—such as multi-utility representations (Evren & Ok, 2011) and injective monotones (Hack et al., 2022). These alternatives would guarantee transitivity or antisymmetry. Our present formulation, however, intentionally approximates the characteristic function of the underlying partial order, treating the preorder properties more as an inductive bias that encourages hierarchical structure rather than a strictly enforced constraint. We have explicitly mentioned this design choice and provided a discussion of these alternatives as future work.
>
> 3. Handling pairs of trajectories without an inclusion relation: We thank the reviewer for highlighting this point. We have expanded the explanation of our contrastive training setup and added pseudo-code (Algorithm 1) that clearly shows how pairs with and without subset relations are handled. Specifically, pairs where neither trajectory is contained in the other are used as negative samples in the contrastive loss, ensuring that the energy function assigns low compatibility to unrelated or non-hierarchically connected goals.
>
> 4. Potential drift during hierarchy traversal: We have revised the relevant section to clarify how optimization interacts with the energy function. Traversal operates by optimizing over one argument while fixing the other. Maximizing the energy with respect to the second argument yields more abstract goals, whereas maximizing with respect to the first yields more concrete ones. Because the energy is always conditioned on one (or multiple) fixed reference goal(s), the gradient steps are guided toward goals that remain related to this reference rather than drift to arbitrary points in the space. We added further discussion to clarify this mechanism and prevent misunderstanding.
>
> 5. Notation and clarity: We have addressed all notational issues and clarified missing definitions. In particular, the definition of $p_\vartheta$ was accidentally omitted in the original submission. We actually simplified the formula about reward encoding (directly parametrizing $g_\vartheta$). Regarding $E_\theta(\cdot,\cdot)$, we thank the reviewer for noting this inconsistency. It was always meant to denote the same neural network mapping two inputs to a continuous scalar between $0$ and $1$, approximating the characteristic function of the partial order. We have clarified this throughout (e.g. in Eq. 3) and improved the flow and readability.
>
> 6. Baseline: We expanded the experimental section to include an additional comparison with a VAE-based approach. The revised results show that our approach improves the results on abstract goal tasks, in particular in terms of representing unseen reward functions through single abstract goals.

---

### Official Review · Reviewer_Dy4d · 2025-10-31

**Soundness:** 2
**Presentation:** 1
**Contribution:** 2
**Rating:** 4
**Confidence:** 3

**Summary:**

The paper introduces an algorithm for integrating concrete and abstract goals into a single shared latent goal space. This space is learned using asymmetric contrastive learning, which encourages the representations of related concrete and abstract goals to be similar, which induces a subset relation that enables traversing the latent abstraction to control the level of goal abstraction.

Concrete goals are defined as single observations, whereas abstract goals are sequences of observations. These sequences are encoded using an RNN, and contrastive learning is applied to the resulting representations. Positive pairs consist of related sequences (specifically, when one sequence is a subset of another), while negative pairs consist of unrelated sequences or the inverse of the positive pairs.
The paper suggests an optimization-based method to traverse the abstraction level in the latent space, based on gradinet from the contrastive energy function

The paper further demonstrates that the learned latent goal space can be used to encode a reward function that depends on the observations. The encoded reward can then be provided to the policy in the form of an abstract goal representation.

**Strengths:**

- The idea of encoding concrete and abstract goals is really interesting.
- Encoding reward in an abstract goal space seems to be a promising approach for extending goal-reaching beyond reaching specific goals in the original state/observation space.

**Weaknesses:**

- The paper is hard to follow, and the lack of a background section makes understanding it even more challenging.
- Coverage assumption is limiting.
- Lack of baselines, one possible baseline is to train some density model over the diverse data, then sample from that model in proportion to the reward.

**Questions:**

- The complete problem setup is still not entirely clear to me. Could you clarify the setup further? Is the abstract goal model trained offline on diverse data and then applied to downstream goal-reaching tasks? I recommend adding a background section that introduces the problem statement and provides the necessary context to understand the rest of the paper.

- For the positive pairs, what do you mean by “as well as composite trajectories formed from unrelated segments” (line 180)? Could you explain this using the notation introduced in the paper? Does this mean that you compose trajectories and then take subsets of them to construct a positive pair? How are these trajectories composed? Could you elaborate on this process?

- How exactly is the relabeling with abstract goals performed? Do you relabel only a subset of the sampled single-observation goals, or are all of them relabeled?

Overall, I like the idea of the paper, but the writing quality needs improvement to make the paper clearer and less confusing, and at lease one strong baseline should be added.

---

> ### Author Response · Authors · 2025-11-20
>
> We would like to thank the reviewer for the valuable feedback, especially noting where background and explanations were unclear. This feedback helped us tighten the structure and improve the clarity of the manuscript. Our responses and the corresponding changes are given below.
>
> 1. Missing background section: We have reorganized the introduction by adding explanatory background information.
> 2. Problem statement: We have added a clear problem statement in the introduction.
> 3. Training phases: We have added an overview of our training phases at the beginning of Section 3. In particular, the agent and its goal representation are learned through environment interactions and are later frozen for evaluation.
> 4. Coverage assumption: During pre-training, we do not assume full coverage of the environment. We have updated the analysis presented in Figures 6, 8, and 9. These results, which were previously based on an exhaustive dataset assuming full environmental coverage, are now performed on data obtained directly through environment interactions. To generate the heatmaps, we sample observations from the environment with broad coverage to probe the representation (just for visualization) but we never train on this data.
> 5. Contrastive learning: We have added pseudo-code to clarify the use of positive and negative pairs in our approach.
> 6. Relabeling with abstract goals: We sample the length of the sequence that is encoded as the relabelled goal once per batch, without generating more data. We have clarified this in the corresponding section.

---

> ### Comment · Reviewer_Dy4d · 2025-11-23
>
> Dear Authors,
>
> Thank you for the response and the clarifications.
>
> The introduction is much clearer, and I really appreciate the reorganization.
>
> The pseudo-code clarifies the contrastive learning procedure, but now that I understand it better, one question on my mind is how do you ensure that the contrastive model satisfies the subset relation? Are you assuming the contrastive model will learn it just from the positive and negative pairs? Can you be a bit more formal?
>
> The notation of the encoding reward function (line 232) is not clear. What exactly is
> ? Can you clarify? And at the post-training phase, do you provide this $g_{\mathbb{\vartheta}} as an input (goal) to the policy?
>
> Best.

---

> > ### Author Response · Authors · 2025-11-24
> >
> > We appreciate the reviewer’s additional feedback and the opportunity to further clarify our work. Answers to the specific questions are provided below.
> >
> > **Question:**
> >
> > > The pseudo-code clarifies the contrastive learning procedure, but now that I understand it better, one question on my mind is how do you ensure that the contrastive model satisfies the subset relation? Are you assuming the contrastive model will learn it just from the positive and negative pairs? Can you be a bit more formal?
> >
> > **Answer:**
> >
> > In the revised version, we analyze to what degree the subset relation is fulfilled by the energy function. In particular, we analyze basic transitive properties of the learned relation in the appendix (A.1.3), indicating that the relation is approximately transitive, in line with the subset requirement. This suggests that with enough data, our contrastive learning approach is able to capture at least a soft subset relation without imposing additional inductive biases such as explicitly enforcing transitivity. The advantage of not having a strict binary relation but an energy function that induces a soft relation is that it allows for:
> > - comparison between trajectories which are not necessarily in a strict subset relation (e.g. slightly non-overlapping trajectories) and
> > - optimization with gradient ascent, where we maximize the energy function in order to find more concrete/more abstract goals (Eq. 1).
> >
> >
> > We have clarified this in the manuscript.
> >
> > **Question:**
> > > The notation of the encoding reward function (line 232) is not clear. What exactly is ? Can you clarify? And at the post-training phase, do you provide this $g_{\mathbb{\vartheta}} as an input (goal) to the policy?
> >
> > **Answer:**
> >
> > We thank the reviewer for pointing this out. We clarify that the reward encoding mechanism is used exclusively during the evaluation phase. Specifically, it is used for the persuadability analysis (Figures 7 and 10) and to verify the goal representation's capability to capture novel rewards (Figures 6, 8, and 9). In practice, we translate rewards into goals by optimizing for a $\mathbf{g}_{\vartheta}$ that maximizes the respective reward (Eq. 3). For the persuadability analysis, we input this optimized goal into the policy and analyze the resulting behavior.
> > We have edited our manuscript by merging the two relevant sections as "Generalization to novel rewards". Additionally, we have changed the notation in Eq. 3 for better readability.

---

> > > ### Comment · Reviewer_Dy4d · 2025-11-26
> > >
> > > Dear authors
> > >
> > > Thank you for the continuing discussion.
> > >
> > >  > learned relation in the appendix (A.1.3)
> > >
> > > I see that the abstraction steps for the analysis are up to 6. I understand that it may become a difficult optimization problem for a large number of abstraction steps. What was the maximum sequence length used in training the contrastive model? I know it was sampled randomly, but what was the maximum value, and what distribution did you use for sampling the sequence length?
> > >
> > > Best

---

> > > > ### Author Response · Authors · 2025-11-27
> > > >
> > > > Thank you as well for your helpful comments.
> > > >
> > > >
> > > > - **Abstraction steps**: The number of steps in A.1.3 are not directly related to the sequence length used during training. More specifically, for each abstaction step we maximize the energy with $50$ gradient ascent steps (with the intent to find a goal with energy = 1) using Eq. 1. For our proposed hindsight training method with abstract goals (see Sec.2, 'Abstract hindsight relabeling'), we perform one abstraction step by maximizing Eq. 1.
> > > >
> > > > - **Sequence lengths**: During training, we first uniformly sample trajectories of length $T \sim \text{Unif}(3,50)$ and split them up into positive and negative sub-trajectories for contrastive learning. For each sub-trajectory we sample the length independently from $\text{Unif}(1,\lfloor T/3 \rfloor)$. This way we can guarantee that for each full trajectory of length $T$ we can find two non-overlapping sub-trajectories of at most length $\lfloor T/3 \rfloor$ with a gap of at most $\lfloor T/3 \rfloor$ steps in between. We have added this information in Appendix A.3.
> > > >
> > > > - **Transitivity**: Regarding the decrease in transitivity over multiple abstraction steps: Abstractions are created during training based on the experienced observation sequences. This means, that we do not expect to find a hierarchy of arbitrary supersets of goals, but rather one that is grounded in the agent's experience.
> > > >
> > > >     We have clarified this in the paper as follows:
> > > >
> > > > > Starting from a goal $g_0$ grounded in the agent's experience, we generate a sequence $(g_0,g_1,\dots,g_t)$ of goals iteratively. Subsequent goals are obtained through optimization of the energy following Equation 1, i.e., $g_t = \arg\max_g E_{\theta}(g_{t-1},g)$. Consequently, subsequent goals do not necessarily correspond to a particular experienced sequence but, for example, might represent a disjoint set of observations that share some common feature (e.g., being close to a wall, center of a room, corners, etc.). Since the energy never experienced these more abstract goals during training, we naturally expect a degradation in goal quality as more steps are taken to obtain more and more abstract goals. We see this degradation directly in the decreasing final energy values achieved through this optimization (Fig. 11d), which directly matches the decrease in transitivity (Fig. 11a).

---

### Official Review · Reviewer_xfwQ · 2025-11-04

**Soundness:** 2
**Presentation:** 2
**Contribution:** 2
**Rating:** 6
**Confidence:** 3

**Summary:**

The authors investigate goal learning in reinforcement learning, with a focus on distinguishing concrete goals (specific target states) from abstract goals (higher-level goal concepts that subsume many concrete target states). They propose a self-supervised method for learning abstract goals from trajectories aimed at concrete goals, enabling an agent to reason at two complementary levels: concrete (state-specific) and abstract (more conceptual). The key idea is that trajectories implicitly contain a successor structure, some states reliably precede and enable others, and by encoding these successor relations the agent can induce a partial order or some hierarchical structure over goals. A learned sequence-to-goal encoder maps states into a latent space where abstract goals emerge as representations that generalize across multiple concrete goal outcomes.

**Strengths:**

The authors evaluate their approach across three environments, GridWorld, MemoryMaze, and FetchPush, and assess both (i) the ability to represent diverse reward functions via abstract goal embeddings, and (ii) generalization to novel tasks using abstract goal targets for control. Results show that the proposed method better captures spatial and compositional structure in reward functions compared to baselines that operate only on concrete/observation-level goals, and exhibits improved performance when transferring to new tasks.

**Weaknesses:**

The paper offers an interesting perspective but there are some observations and questions:

- The method assumes access to complete state trajectories under full observability during self-supervised training. It would be valuable to discuss how the approach could be adapted to partial observability (e.g., via belief states, recurrent encoders, or trajectory snippets). Moreover,  constructing and organizing abstractions from full trajectories may become prohibitively expensive as state/action dimensionality and horizon grow. The paper would benefit from an analysis (or bounds) of computational/memory complexity, plus experiments or ablations that probe scaling behavior.
- The paper argues that abstract goals enable more conceptual reasoning than concrete goals; however, the quality and correctness of these abstractions are difficult to assess. Since the abstraction emerges implicitly from trajectories, there is no principled mechanism to verify whether a learned abstract goal is semantically meaningful, sufficiently general, or overly coarse/incorrect. This raises concerns about reliability and fidelity of the abstraction layer: an abstract goal might oversimplify, collapse distinct states, or fail to correspond to a useful higher-level concept, and the current evaluation does not formally quantify “degree of abstraction” or guarantee abstraction soundness.
- The current evaluation is limited to single-agent, fully observable environments. As a forward-looking direction, the authors could outline how their abstraction mechanism would scale to multi-agent settings, for instance by learning abstractions over joint trajectories, factoring goals across agents (shared vs. individual abstract goals), and adapting successor-based relations to account for other agents’ policies. Even a short roadmap would clarify feasibility and potential challenges.

**Questions:**

- Given the recent progress in LLM-driven abstraction and hierarchical reasoning, it would be interesting to discuss whether the learned abstract goal structure could be compared against abstractions produced by a language-conditioned model. In particular, once the agent induces a successor-based goal hierarchy, could a pretrained LLM (after some light fine-tuning or prompting) derive comparable abstract goal concepts from trajectory descriptions? Such a comparison, even qualitative, could help assess whether the latent hierarchy aligns with human-interpretable abstractions and might provide a complementary validation signal for abstraction quality. While this suggestion may not be perfectly aligned with your current scope, I share it to motivate a potentially fruitful direction: contrasting your latent goal hierarchy with LLM-derived abstractions could be a compelling exploratory angle for future work.

---

> ### Author Response · Authors · 2025-11-20
>
> We thank the reviewer for the helpful comments and perspectives. We address each point below.
>
> 1. Observability and scaling: We do not assume full observability. We only assume the agent receives a sensor reading (observation) each moment in time. The question of scaling is interesting, but can probably only be answered through experiments with extensive computational resources, because the scaling would critically depend on the generalization properties of the energy function, which is notoriously hard to study theoretically. As this paper focusses on the conceptual foundations of this approach, we see this more as a question for future work.
>
> 2. Quality of abstractions: We have added an evaluation that tests our learned energy and abstraction models for consistency regarding transitivity and coverage across a number of gradient ascent applications. In terms of semantics, the difficulty is that there is no independent ground truth that allows to assess the meaning of a particular latent goal. The only two things we can check is whether goals are useful for generalization in the policy (see evaluation) and the internal consistency of the energy function, for which we have added the test mentioned above.
>
> 3. Multi-agent learning: The question how goals would be learned and communicated between different agents is intriguing. As multi-agent learning is not the focus of the current study, we think this could be future work covering dynamic environments and multi-agent scenarios.
>
> 4. Evaluation of latent goals using other models (e.g. LLMs): Currently, we use heatmaps in order to make the latent goals accessible to our human semantics. The reviewer's proposal to use multi-model language models to translate goal-induced behavior into language descriptions would be an interesting endeavor. One particular approach could be to feed the LLM trajectories corresponding to goals with a varying abstraction level, obtained through optimizing Equation 1, and analyzing the change in the textual descriptions. However, we agree with the reviewer that this type of analysis is out of scope of this paper, but we are thankful for this interesting perspective.

---

### Author Response · Authors · 2025-11-20

We sincerely thank all reviewers for their constructive comments, which guided the revisions made to the manuscript. In addition to our comments below, we have marked all changes directly in the manuscript. The code showcasing our approach is now available on github: https://github.com/concretetoabstract2025/concretetoabstract/

---

### Author Response · Authors · 2025-12-02

We thank all reviewers for their constructive feedback which helped us to improve the manuscript not only in presentation, but also in substance.  All changes since the initial submission are highlighted in magenta in the revised version of our paper. We summarize the issues raised by the reviewers as well as our changes to the manuscript in the following.

### **(1) Major comments that led to substantial changes**
  - **Core theory not well motivated** (xfwQ, Xcv6): We acknowledge that we have conflated two notions of abstraction in the original exposition and we are very grateful to the reviewers for pointing this out. In the revised version we clearly separate the more foundational idea of abstraction representation through subsets from the more applied idea of representing spatial abstractions by subsets of contingent trajectory snippets.

  - **Theoretical justification missing** (eU52): We have added a section on the theoretical underpinnings of representing partial orders, where a representation based on an energy function that approximates the charactersitic function allows representing soft partial orders. We have also added experiments that quantify the extent to which the partial order of the subset is respected by a learned energy function.

  - **Adequate baseline comparison missing** (Dy4d, eU52, Xcv6): In the first submission we only compared to methods based on direct observations. We have added a VAE baseline in the experimental section that demonstrates the value of our approach holds also in comparison to other latent goal representations without hierarchical subset ordering.



### **(2) Major comments based on misunderstandings**
  - **Full observability** (xfwQ): Our paper does not presume full observability as stated by one of the reviewers. We only assume the agent receives a sensor reading each moment in time, which is typically a partial observation (e.g. first person images, local lidar signals, etc.), depending on the environment.
  - **Full coverage** (Dy4d): Despite being claimed by one reviewer, our paper does not require full coverage of the state space during pre-training. For our evaluation, we have used data uniformly covering the environment, but the agent is never trained on this data. We have clarified this in the paper.
  - **Hierarchy traversal drift** (eU52): Our approach does not suffer from a problem of potential drift during hierarchy traversal as stated by one of the reviewers that falsly assumed that the traversal is achieved by optimization of a single argument of the energy function, which would indeed cause a problem in one direction. However, we optimize either the first or the second argument depending on which direction we want to traverse the hierarchy. Thus, this problem does not occur.


### **(3) Improvements to presentation**
  In addition, the reviewers' comments helped us to improve the presentation of our manuscript. We have added a much more detailed background section in the introduction, a clear problem statement, a clearer separation of foundational theory from experimental application, pseudocode in the Methods, clearly structured details for the training and testing phases, as well as an adapted conclusion.

---

### Meta-Review · Area_Chair_5G7R · 2026-01-07

**Summary:**

The paper proposes a hierarchical goal representation for self-supervised reinforcement learning that integrates concrete and abstract goals into a unified latent space. This framework aims to enable agents to generalize across tasks by navigating between concrete and abstract goals. Despite the interesting conceptual premise, the submission was met with skepticism regarding its theoretical foundations and practical execution. While the authors provided a significant revision during the rebuttal fundamental concerns persisted.

**Reviewer Concerns:**

- The authors successfully addressed the lack of baselines by adding a VAE comparison. They also clarified misunderstandings regarding full observability and coverage assumptions.

- The core premise remains a sticking point. Reviewer Xcv6 noted that equating "longer trajectory" with "more abstract goal" is a heuristic that fails in many contexts. The rebuttal's clarification that this is just "one kind of abstraction" did not resolve the concern that the paper's primary mechanism relies on a potentially flawed proxy.

- Reviewer eU52 raised valid concerns about the theoretical justification for using energy functions to model asymmetric partial orders. The authors' defense was viewed as an implementation detail rather than a rigorous theoretical guarantee of satisfying partial order properties.

- The disconnect between representation and policy persists.

**Reviewer Scores:**

Reviewer eU52, xfwQ, Xcv6 are unlikely to change.

Reviewer Dy4d might increase the rating but is unlikely to change significantly. While the reviewer appreciates the clarifications, the overall complexity relative to the modest gains remains unresolved.

---

### Decision · Program_Chairs · 2026-01-26

Reject